# Protective Effects of Hesperetin on Cardiomyocyte Integrity and Cytoskeletal Stability in a Murine Model of Epirubicin-Induced Cardiotoxicity: A Histopathological Study

**Adina Pop Moldovan** [1,†] , **Simona Dumitra** [2,†] , **Cristina Popescu** [3,†] , **Radu Lala** [1] , **Nicoleta Zurbau Anghel** [4] , **Daniel Nisulescu** [4] , **Ariana Nicoras** [4] , **Coralia Cotoraci** [5] , **Monica Puticiu** [6,*] , **Anca Hermenean** [4,*] and **Daniela Teodora Marti** [7]

[1] Department of Cardiology, Faculty of Medicine, Vasile Goldis Western University of Arad, 310025 Arad, Romania; pop-moldovan.adina@uvvg.ro (A.P.M.); lala.radu@uvvg.ro (R.L.)

[2] Department of Pediatrics, Faculty of Medicine, Vasile Goldis Western University of Arad, 310025 Arad, Romania; dumitra.simona@uvvg.ro

[3] Department of Genetics, Faculty of Medicine, Vasile Goldis Western University of Arad, 310025 Arad, Romania; popescu.cristina@uvvg.ro

[4] Department of Histology, Faculty of Medicine, Vasile Goldis Western University of Arad, 310025 Arad, Romania; zurbau-anghel.nicoleta@uvvg.ro (N.Z.A.); nisulescu.daniel-dumitru@uvvg.ro (D.N.); nicoras.violeta-ariana@uvvg.ro (A.N.)

[5] Department of Hematology, Faculty of Medicine, Vasile Goldis Western University of Arad, 310025 Arad, Romania; cotoraci.coralia@uvvg.ro

[6] Department of Emergency, Faculty of Medicine, Vasile Goldis Western University of Arad, 310025 Arad, Romania

[7] Department of Microbiology, Faculty of Medicine, Vasile Goldis Western University of Arad, 310025 Arad, Romania; marti.teodora@uvvg.ro

\* Correspondence: puticiu.monica@uvvg.ro (M.P.); hermenean.anca@uvvg.ro (A.H.)

† These authors contributed equally to this work.

**Abstract:** Anthracyclines, including epirubicin (Epi), are effective chemotherapeutics but are known for their cardiotoxic side effects, primarily inducing cardiomyocyte apoptosis. This study investigates the protective role of hesperetin (HSP) against cardiomyopathy triggered by Epi in a murine model. Male CD1 mice were divided into four groups, with the Epi group receiving a cumulative dose of 12 mg/kg intraperitoneally, reflecting a clinically relevant dosage. The co-treatment group received 100 mg/kg of HSP daily for 13 days. After the treatment period, mice were euthanized, and heart tissues were collected for histopathological, immunofluorescence/immunohistochemistry, and transmission electron microscopy (TEM) analyses. Histologically, Epi treatment led to cytoplasmic vacuolization, myofibril loss, and fiber disarray, while co-treatment with HSP preserved cardiac structure. Immunofluorescent analysis of Bcl-2 family proteins revealed Epi-induced upregulation of the pro-apoptotic protein Bax and a decrease in anti-apoptotic Bcl-2, which HSP treatment reversed. TEM observations confirmed the preservation of mitochondrial ultrastructure with HSP treatment. Moreover, in situ detection of DNA fragmentation highlighted a decrease in apoptotic nuclei with HSP treatment. In conclusion, HSP demonstrates a protective effect against Epi-induced cardiac injury and apoptosis, suggesting its potential as an adjunctive therapy in anthracycline-induced cardiomyopathy. Further studies, including chronic cardiotoxicity models and clinical trials, are warranted to optimize its therapeutic application in Epi-related cardiac dysfunction.

**Keywords:** epirubicin; hesperetin; cardiac toxicity

## 1. Introduction

Anthracyclines rank as some of the most potent chemotherapy drugs, crucial for treating a range of adult cancers such as breast cancer, sarcomas, and lymphomas. Their anticancer efficacy is due in part to their ability to intercalate into nuclear DNA, inhibit

topoisomerase II, and generate reactive oxygen species (ROS), leading to widespread cellular damage including DNA, mitochondria, and cell membranes, and triggering apoptosis in cardiomyocytes [1]. Despite their efficacy, the clinical application of anthracyclines is curtailed by cardiotoxicity that is proportional to the dose administered, with lower doses potentially compromising the response to cancer treatment.

Apoptosis of cardiomyocytes is a critical aspect of the cardiac damage induced by anthracyclines. Recent studies underline the significant role of this apoptosis in the development of drug-induced cardiomyopathy [2–4]. To counteract the oxidative stress provoked by anthracycline medications, the use of antioxidants, whether natural or synthetic, has been proposed. Agents like N-acetylcysteine, berberine, and silymarin have shown promise in biochemically and histologically preventing doxorubicin-induced cardiotoxicity in rat models [5–7]. Similarly, melatonin has been observed to safeguard heart tissue from epirubicin-induced toxicity [8].

Hesperetin (HSP) is a flavanone, a type of flavonoid, which is a class of compounds with antioxidant effects found naturally in a variety of plants. HSP is particularly abundant in citrus fruits and is the predominant flavanone in oranges and lemons. It is the aglycone form of hesperidin, which means that hesperetin is hesperidin without its sugar part. HSP, like other flavonoids, is known for its potential health benefits. It has been studied for its antioxidant properties, which can help to protect cells from damage caused by free radicals [9–11]. This action is the basis for its potential role in reducing inflammation and fighting against various diseases, including cancer and cardiovascular diseases.

HSP is known for its broad range of pharmacological benefits, including its role as an antioxidant [9,10,12], anti-inflammatory [13], anticarcinogenic [14], and neuroprotective [15,16] agent. While prior research has shown that HSP can mitigate the cardiotoxicity induced by doxorubicin by reducing oxidative stress [17], the specific effects of HSP on cardiomyocyte apoptosis and desmin integrity in the context of epirubicin-induced damage have not yet been explored.

The study hypothesizes that HSP has cardioprotective properties capable of reducing the cardiotoxic effects of epirubicin (Epi), a chemotherapeutic agent known for induction apoptosis in cardiomyocytes, leading to cardiomyopathy. Through histopathological analysis, along with immunofluorescence, immunohistochemistry, and transmission electron microscopy (TEM), the study aims to evaluate whether HSP treatment preserves cardiac morphology, modulates the expression of apoptosis-regulating Bcl-2 family proteins, and maintains mitochondrial and DNA integrity in cardiac cells. This histopathological approach seeks to establish the efficacy of HSP in preventing the characteristic cardiac damage associated with Epi exposure.

## 2. Materials and Methods

### 2.1. Chemicals

HSP was purchased from Sigma-Aldrich (St. Louis, MO, USA) and Actavis epirubicin 2 mg/mL (Davie, FL, USA).

### 2.2. Animals

Adult male CD1 mice were maintained on a standard rodent diet and experienced a consistent 12 h light/dark cycle. The environment was carefully regulated at a temperature of approximately 23 °C and a humidity level of around 50%. The mice had free access to food and water throughout the study. All procedures involving the animals were conducted following ethical guidelines and were authorized by the Research Ethical Committee of Vasile Goldis Western University of Arad.

### 2.3. Experimental Design

The study included four groups of animals, with each group containing 10 subjects:

1. Control group—received a daily oral administration of 0.7% carboxymethyl cellulose (CMC) solution for 13 consecutive days.

2.   Epi group—received a total of 12 mg/kg Epi (cumulative dose consistent with human clinical treatments), administered intraperitoneally. This total dose was divided into six equal parts, with each part being 2 mg/kg and given every other day, starting from the second day of the experiment.

3.   Epi+HSP group—each animal was given 100 mg/kg of HSP orally (dissolved in a 0.7% CMC solution), starting one day before the initial Epi injection and continuing daily up to 13th day of the experiment.

4.   HSP group—received a daily oral administration of HSP in 0.7% carboxymethyl cellulose solution, like group 3.

The selected dosage of HSP at 100 mg/kg was determined based on previous dose–response research that highlights its effectiveness in providing cardioprotective benefits [18].

On the 14th day, all mice were anesthetized by inhalation of isoflurane and euthanized by cervical dislocation. Heart samples were preserved in a buffered formalin solution to obtain histological sections.

### 2.4. Histopathology

Heart tissue samples were processed by fixation in 4% formaldehyde solution in PBS, followed by paraffin embedding. Sections of 5 μm thickness were prepared and subjected to staining with hematoxylin and eosin and Fouchet van Gieson's stain as per Bio-Optica staining kit protocols. Structural evaluations of the sections were conducted using an Olympus BX43 light microscope, and the images were captured with XC30 imaging software.

### 2.5. Immunofluoresce

The immunofluorescence protocol was initiated with deparaffinization of paraffin-embedded heart sections, followed by rehydration in graduated ethanol solutions of 100%, 96%, and 70%. Post-rehydration, the sections were washed and subjected to antigen retrieval in a sodium citrate buffer, pH 6.0. To prevent non-specific antibody binding, the sections were treated with a 5% normal goat serum blocking solution for an hour. Afterwards, the slides were incubated with primary antibodies specific for Bax and Bcl-2 (Santa Cruz Biotechnology Inc., Dallas, TX, USA) at a dilution of 1:100. This incubation was carried out overnight at 4 °C. Following this, the slides were washed and then incubated with secondary antibodies matched to the primary antibodies at a 1:100 dilution for one hour at room temperature. Nuclei were stained with DAPI, and the slides were preserved in a fluorescent mounting medium for observation using a Leica TCS SP8 confocal microscope (Wetzlar, Germany).

Quantitative assessment of Bax and Bcl-2 fluorescent markers was performed with Image J 6.4 software. This involved measuring fluorescence intensity in multiple regions per section. For each heart section, five random fields were chosen for analysis. The resultant data were presented as a percentage of fluorescence intensity relative to the control group, with the control's intensity normalized to 100%.

### 2.6. Immunohistochemistry

Sections with a thickness of 5 μm were processed by removing paraffin with toluene and then subjected to a rehydration sequence. Antigen retrieval was carried out using Novocastra solution (Leica Biosystems, Nußloch, Germany). The tissue sections were then exposed to a 3% hydrogen peroxide solution to suppress the activity of endogenous peroxidase enzymes for 10 min. Following this, Novocastra blocking solution was applied to prevent non-specific antibody binding for another 10 min. The sections were then incubated with primary antibodies targeting desmin and caspase-3 at a dilution ratio of 1:100, in a 4 °C environment overnight. For the visualization of these antibodies, the Novolink Max Polymer Detection System was utilized, employing 3,3′-diaminobenzidine (DAB) as the chromogen. To highlight the nuclei, hematoxylin staining was performed. The

final step involved dehydrating the sections and preparing them with a mounting medium for analysis. Microscopic imaging was performed using an Olympus BX43 system.

### 2.7. Transmission Electron Microscopy (TEM)

The heart tissue specimens were first preserved in a 2.7% glutaraldehyde solution mixed in 0.1 M phosphate buffer and maintained at 4 °C for 90 min. The samples were then cleansed with a 0.15 M phosphate buffer, pH adjusted to 7.2, and subsequently further fixed in a 2% osmium tetroxide solution, which was also prepared in 0.15 M phosphate buffer, for an additional hour at 4 °C. The dehydration of the samples was achieved through a graduated series of acetone immersions, after which they were embedded in Epon 812, a type of epoxy embedding resin. Ultrathin sections with a thickness of 60 nm were then produced using a Leica EM UC7 ultramicrotome (Wetzlar, Germany). These sections were contrasted with uranyl acetate and lead citrate stains for enhanced visualization and analyzed under a Tecnai 12 Biotwin transmission electron microscope.

### 2.8. In Situ Detection of DNA Fragmentation

The process for detecting nuclear DNA fragmentation within the tissue sections utilized the In Situ Apoptosis Detection Kit from Calbiochem (EMD Chemicals, San Diego, CA, USA). The method employed was the terminal deoxynucleotidyl transferase (TdT)-mediated dUTP Nick End Labeling (TUNEL) assay, which was performed according to the manufacturer's protocol. This assay incorporates biotinylated nucleotides into fragmented DNA, which is then detected by binding to streptavidin-horseradish peroxidase (HRP). A reaction with diaminobenzidine (DAB) results in the formation of a dark brown precipitate at the DNA fragmentation sites. To counterstain the nuclei, a methyl green solution provided with the kit was used.

For each examined tissue section, TUNEL-positive nuclei were tallied in ten separate microscope fields. Imaging was conducted using an Olympus BX43 light microscope (Shinjuku, Tokyo, Japan), and digital analysis was performed with Image J software version no. 1.8.0. The apoptotic index was determined by calculating the ratio of TUNEL-positive nuclei to the total number of nuclei, which had been previously stained with Nuclear Fast Red.

### 2.9. Statistical Analyses

Analyses of the data were carried out using GraphPad Prism version 6.0. The results are presented as the mean $\pm$ standard deviation (SD). For statistical analysis, a one-way analysis of variance (ANOVA) was used, with adjustments made for multiple comparisons via the Bonferroni correction method. A $p$-value less than 0.05 was considered to indicate statistical significance.

## 3. Results

### 3.1. Hesperetin Protects and Maintains the Structural and Ultrastructural Integrity of Cardiac Tissue Affected by Epirubicin-Induced Damage

Heart tissue sections from the control group and those treated with HSP displayed typical cardiac histology. However, heart cells in the group treated with Epi exhibited changes such as cytoplasmic vacuolization, myofibril loss, and disarray of fibers (Figure 1). Additionally, light microscopy identified inflammatory cell infiltration and a slight increase in interstitial collagen fibers. The heart tissue of the group receiving both treatments largely resembled that of the control group.

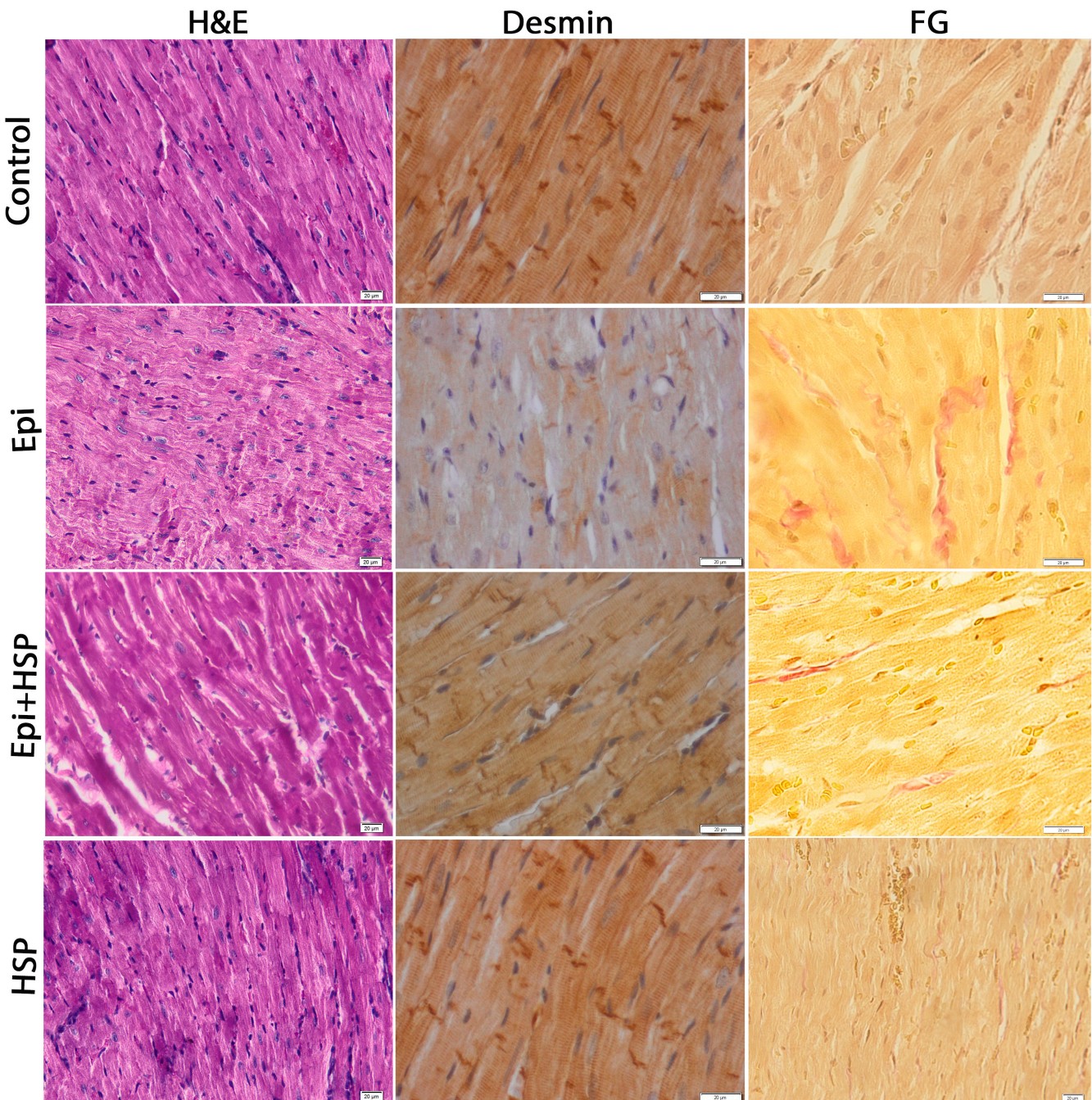

**Figure 1.** Hesperetin protects and maintains the structural architecture of the cardiac tissue affected by epirubicin-induced damage. H&E—hematoxilin and eosin stain; FG—Fouchet van Gieson stain (collagen-red); immunohistochemical (IHC) expression and specific cardiac distribution of desmin; magnification ×20 (H&E); ×40 (desmin and FG).

The distribution of desmin, a key intermediate filament crucial for maintaining cell structure, was examined in the context of cardiotoxicity caused by epirubicin. In the control mice, immunostaining for desmin was present in the cardiomyocytes. Following Epi treatment, there was a noticeable disorganization of the intermediate filaments, as indicated by the weakened and sporadic expression of desmin. This contrasted with the control and was shown to be improved by co-treatment with HSP, as shown in Figure 1.

Electron microscopy showed that the control and HSP-treated tissues maintained normal ultrastructure, with mitochondria appearing to be mostly round or polygonal and

containing densely arranged cristae. In contrast, Epi treatment resulted in disruption of normal structure, including vacuolization, myofibrillar degradation, mitochondrial cristae loss, and localized accumulation of lipids and collagen (Figure 2). The group that received the combined treatment showed alleviated ultrastructural architecture.

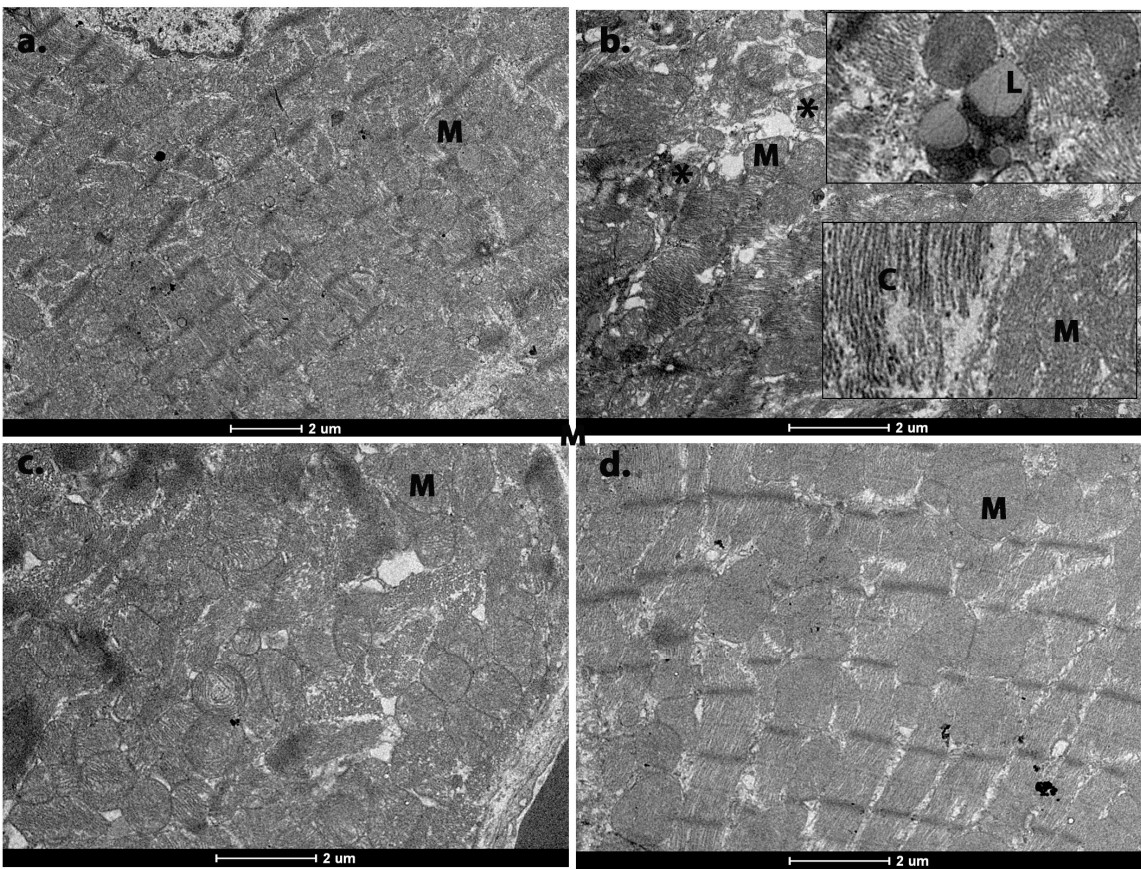

**Figure 2.** Hesperetin protects and maintains the ultrastructural integrity of the cardiac tissue affected by epirubicin-induced damage. (**a**) Control group; (**b**) epirubicin group; (**c**) epirubicin and HSP group; (**d**) HSP group; M—normal aspect of mitochondria; *—mitochondria with rarefied cristae; L—lipids; C—collagen.

*3.2. Hesperetin Provides Protection for Cardiomyocytes against Apoptosis Caused by Epirubicin-Induced Damage*

Additionally, we investigated the impact of Epi on the regulation of Bcl-2 family proteins, which are associated with the mitochondrial membrane. Treatment with Epi markedly upregulated the expression of the pro-apoptotic protein Bax in heart tissue (as shown in Figure 3; $p < 0.001$) and resulted in a decreased expression of the anti-apoptotic protein Bcl-2 (Figure 4). Furthermore, staining for caspase-3 with immunoperoxidase highlighted considerably immunostaining of caspase-3 in the EPI-treated group, indicative of pronounced apoptotic activity in the cardiac tissue (Figure 5) when compared with control tissue ($p < 0.001$). The immunohistochemical pattern was restored after the treatment with flavonoid.

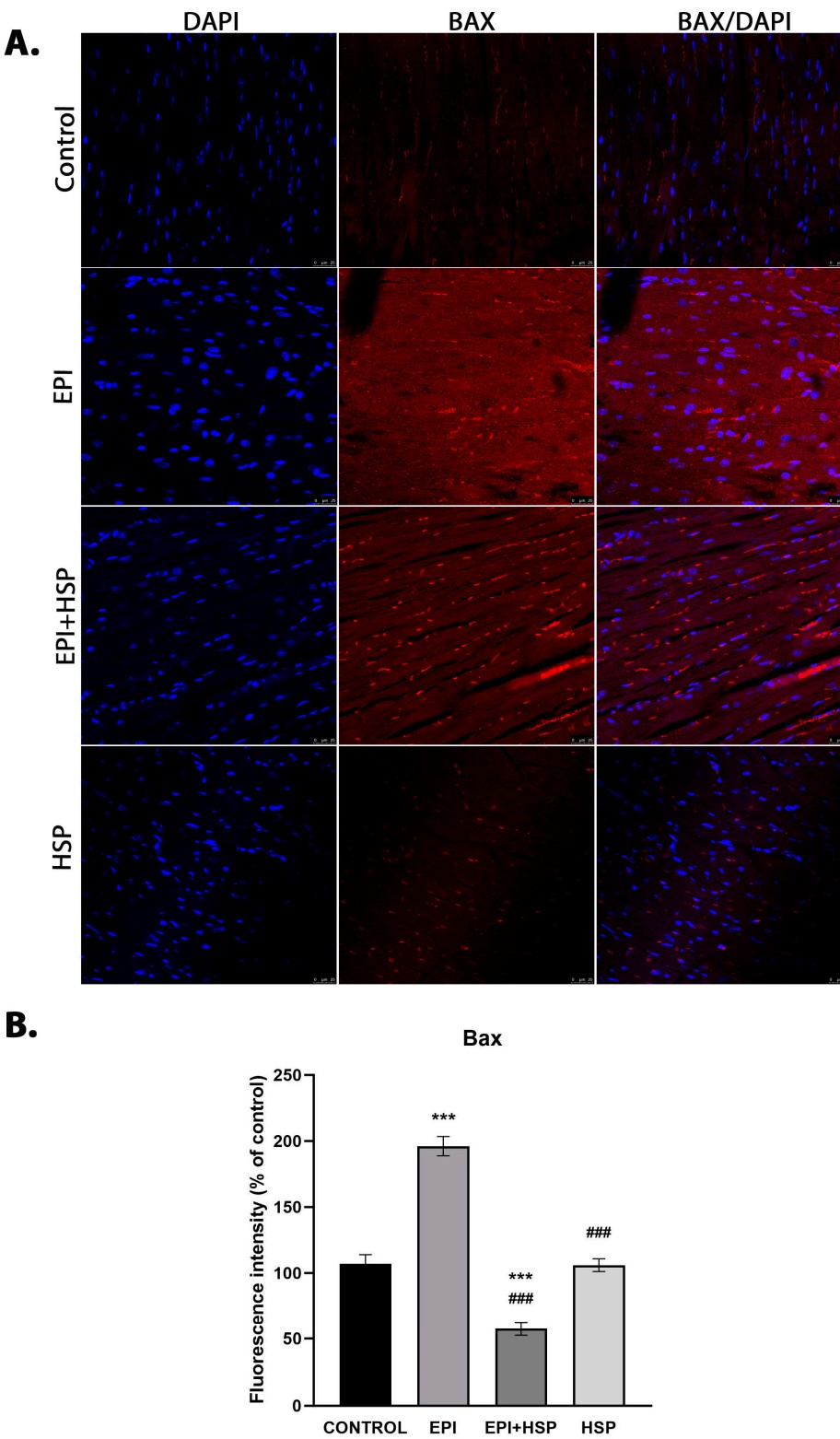

**Figure 3.** (**A**) Expression of BAX in cardiac tissue by immunofluorescence (magnification 62×). (**B**) Bar graphs showing semi-quantification of fluorescence intensity for Bax. *** statistical significance at $p < 0.001$ as compared to control; ### statistical significance at $p < 0.001$ as compared to Epi group.

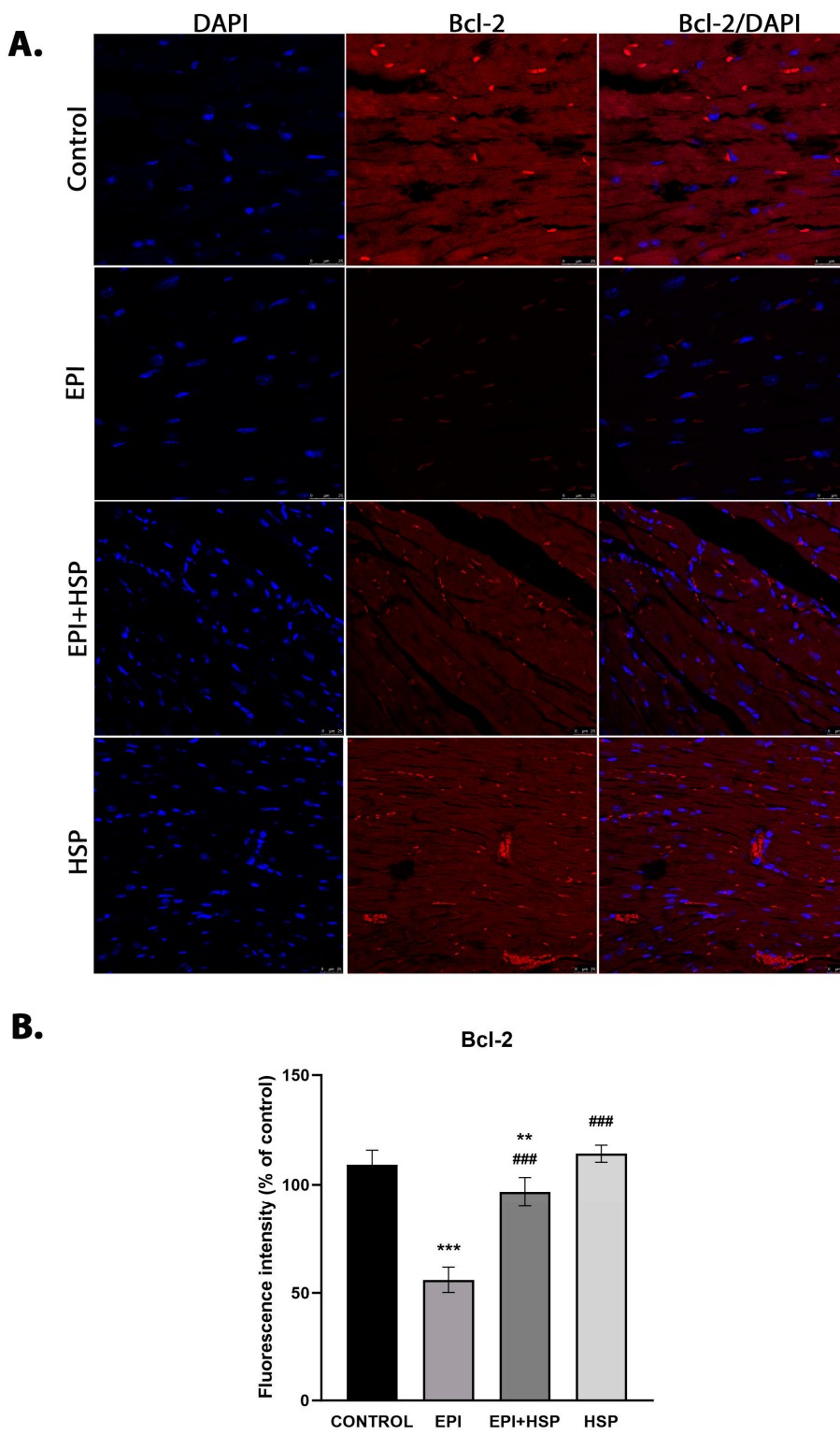

**Figure 4.** (**A**) Expression of Bcl-2 in cardiac tissue as revealed by immunofluorescence (magnification 62×). (**B**) Bar graphs showing semi-quantification of fluorescence intensity for Bcl-2. *** statistical significance at $p < 0.001$ as compared to control; ** statistical significance at $p < 0.01$ as compared to control; ### statistical significance at $p < 0.001$ as compared to Epi group.

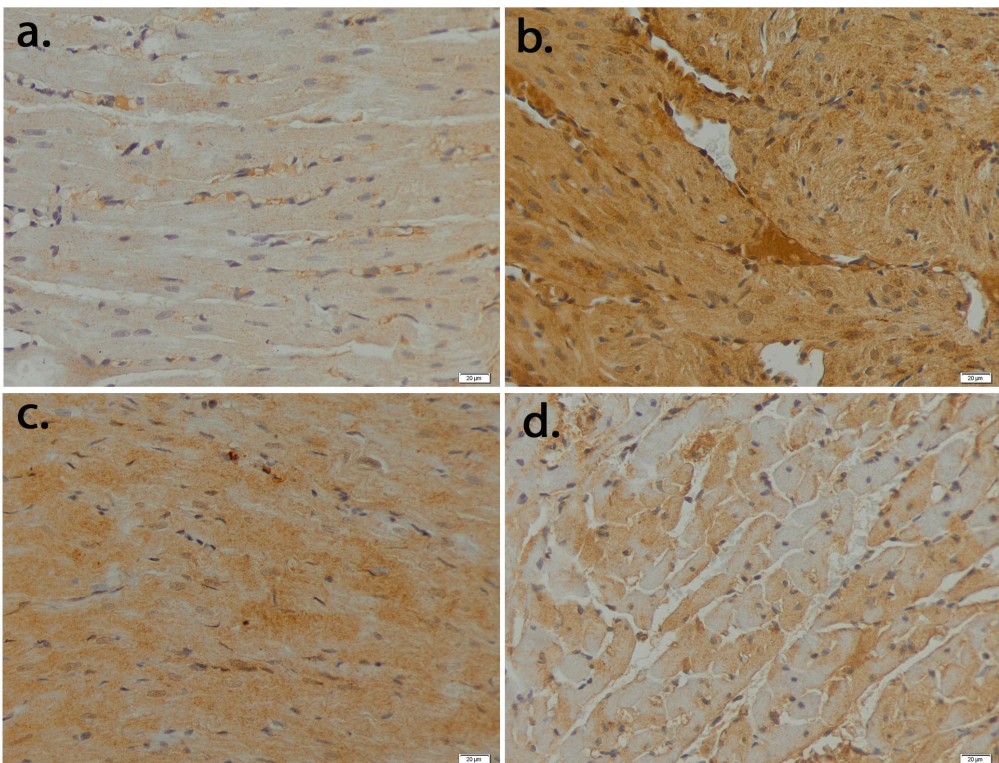

**Figure 5.** The expression and specific cardiac distribution of Caspase-3: (**a**) control; (**b**) epirubicin group; (**c**) epirubicin + hesperetin group; (**d**) hesperetin group; Barr 20 μm.

Epirubicin treatment increased TUNEL-positive nuclei significantly compared to control ($p < 0.001$). The positive nuclei decreased significantly when HSP was co-administered to EPI, compared with EPI alone ($p < 0.001$; Figure 6).

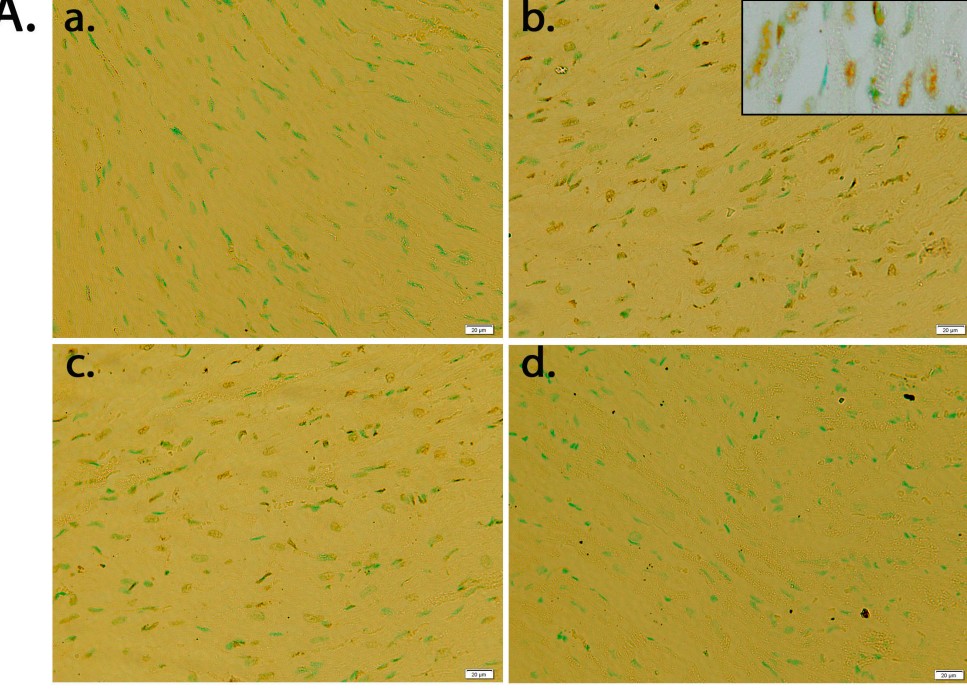

**Figure 6.** *Cont*.

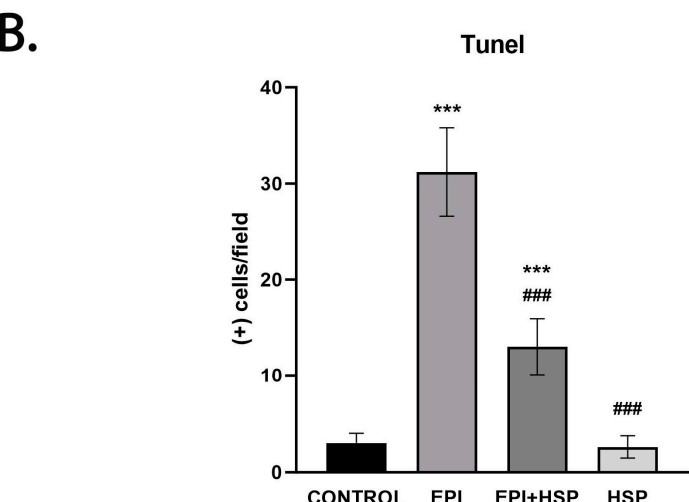

**Figure 6.** (**A**). TUNEL staining of cardiac sections: (**a**) control; (**b**) epirubicin group—most cells exhibit dark-stained TUNEL + nuclei (**c**); epirubicin + hesperetin group—TUNEL-positive cells were significantly decreased and exhibit less darkly stained nuclei than the Epi group; (**d**) hesperetin group. (**B**). Tunel (+) cells/field. A value of $p < 0.01$ was considered significant; \*\*\* as compared to control; ### as compared to Epi group. Magnification 20×.

## 4. Discussion

The cardiac toxicity associated with anthracyclines is characterized by oxidative stress and apoptotic processes. Numerous studies conducted both in vitro and in vivo have suggested that the death of cardiomyocytes due to apoptosis plays a central role in the development of cardiomyopathy caused by anthracyclines [1].

In the current research, the cardiotoxic effects of epirubicin were identified through histological assessments of heart tissue. We noted disturbances in the intermediate filament network and cytoplasmic vacuolization, along with the loss of myofibrils, disorganization of fibers, and a slight increase in interstitial collagen fibers. These findings are consistent with previous observations made for other anthracyclines such as doxorubicin [19] and mitoxantrone [20]. Similar disruptions in cellular organization within the heart have been previously reported in rodent studies following epirubicin treatment [8,21]. Our results showed that co-treatment with hesperetin offers significant preservation of cardiac cell morphology, as evident from the histological analyses.

Mitochondrial degeneration has also been recognized as a key aspect of subcellular damage caused by epirubicin [8,22]. Cardiac mitochondria possess an array of enzymatic defenses such as manganese-superoxide dismutase (Mn-SOD), glutathione reductase, and glutathione peroxidase, which modulate the redox balance in response to drug-induced changes [22]. Additionally, mitochondrial outer membrane permeabilization, a conduit for the release of pro-apoptotic factors like Bax and BAK, has been identified [23]. The released cytochrome c forms the apoptosome complex in the cytosol, triggering the caspase cascade, leading to target cleavage and apoptosis [24]. Conversely, Bax and Bak's effect on mitochondrial outer membrane permeabilization is inhibited by antiapoptotic Bcl-2 proteins, including Bcl-2 and Bcl-xL [25].

In this study, we investigated the influence of hesperetin (HSP) on Bcl-2 family activities in the cardiac tissue of mice administered with a cumulative dose of 12 mg/kg of epirubicin. Our findings indicate that a 100 mg/kg dose of HSP significantly reduced the activity of the pro-apoptotic protein Bax and simultaneously enhanced the activity of the anti-apoptotic protein Bcl-2. These observations suggest that HSP's inhibition of cardiomyocyte apoptosis by epirubicin might be linked to the mitochondrial signaling pathway, a conclusion that aligns with prior research indicating that cardiomyocyte death by epirubicin is mitochondrially induced [26]. The HSP concentration used in our study aligns with doses documented in prior investigations, such as those by Kumar et al. in 2013,

which have been shown to attenuate oxidative stress in rodent models [27]. The reduction in apoptosis and the preservation of mitochondrial integrity observed in our histopathological study upon HSP administration could potentially be attributed to its antioxidant capabilities, suggesting that these effects might underlie the cardioprotective benefits of HSP. This is underscored by the well-known link between mitochondrial dysfunction and oxidative stress.

Anthracyclines trigger cardiomyocyte apoptosis through both extrinsic and intrinsic pathways [28,29]. Our research indicates that HSP markedly inhibits caspase-3 activation, which is typically upregulated by Epi toxicity, suggesting that HSP interferes with the intrinsic pathway of apoptosis.

Desmin is an intermediate filament protein found uniquely in muscle and endothelial cells, with cardiac cells having the highest desmin content. It is pivotal in organizing contraction centers and is also abundant around the nucleus, where it attaches to nuclear components [30].

Our findings reveal significant alterations in desmin distribution in the Epi-treated group compared to controls. In the Epi group, desmin's normal pattern was disrupted, evident from reduced cytoplasmic expression and protein aggregates, echoing findings from other research [3]. This disruption appears to be a consequence of diminished Bcl-2 expression, which fails to prevent intermediate filament network cleavage [31], while the release of caspase-3 fosters desmin degradation [32].

### 5. Conclusions

In summary, targeting the prevention of cardiomyocyte apoptosis presents a potential therapeutic strategy for mitigating doxorubicin (DOX)-induced cardiomyopathy. Our findings suggest that HSP may serve as a complementary treatment agent capable of averting the cardiomyopathy initiated by epirubicin (Epi), whilst maintaining its anti-cancer efficacy. Further research is crucial for fully understanding how HSP could benefit the treatment of heart failure caused by EPI. This should include long-term studies on the cardiotoxic effects of HSP, assessments of cardiac biomarkers in the blood, investigations of the molecular mechanisms, and ultimately, clinical trials. Future studies will aim to explore how HSP might influence the efficacy of Epi in combatting cancer cells. This research could yield critical insights into whether HSP can preserve Epi's therapeutic advantages while also protecting cardiac tissues from damage.

**Author Contributions:** Conceptualization, A.P.M. and A.H.; methodology, A.H.; software, C.P. and D.T.M.; validation, A.H.; formal analysis, A.P.M., R.L., D.N., A.N., S.D., C.C. and D.T.M.; investigation, A.P.M., C.P., R.L., N.Z.A., D.N., A.N., S.D., C.C. and D.T.M.; data curation, A.P.M.; writing—original draft preparation, A.P.M., S.D. and C.P.; writing—review and editing, M.P., A.H. and D.T.M.; supervision, M.P. and A.H.; project administration, A.H. and M.P.; funding acquisition, A.H. All authors have read and agreed to the published version of the manuscript.

**Funding:** This research received no external funding.

**Institutional Review Board Statement:** The animal study protocol was approved by the Ethics Committee of the Vasile Goldis Western University of Arad (Approval no. 32/2.10.2016).

**Informed Consent Statement:** Not applicable.

**Data Availability Statement:** The data presented in this study are available on request from the corresponding author. The data are not publicly available due to privacy.

**Conflicts of Interest:** The authors declare no conflicts of interest.

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
