# Peer review of "Protective Effects of Hesperetin on Cardiomyocyte Integrity and Cytoskeletal Stability in a Murine Model of Epirubicin-Induced Cardiotoxicity: A Histopathological Study"

_applsci, doi:10.3390/app14062560_

Round 1

Reviewer 1 Report

Comments and Suggestions for Authors

More than 50 years have passed since the discovery of anthracyclines. Despite significant progress in the discovery of new anticancer drugs, anthracyclines are still highly effective drugs and are included in many cancer treatment regimens. It has long been known that anthracyclines exhibit precipitated cardiotoxicity, in some cases up to 40%.

Research is still ongoing to find molecules that would reduce the damaging effect of anthracyclines on the myocardium or prevent the initiation of programmed death of cardiomyocytes. A lot of drugs have been tested in experimental studies, but none of them has shown efficacy in clinical practice, and none of them is approved from the point of view of evidence-based medicine.

The article is relevant and opens new possibilities. At the same time, further research is needed.

There are no critical comments to the article.

It is advisable to include biomarkers (troponins, BNP) that are validated and are used as criteria for evaluating effectiveness.

Author Response

Dear Reviewer 1

We would like to express our sincere gratitude for the opportunity to submit our revised manuscript for your consideration. We deeply appreciate the insightful critiques and valuable suggestions provided. We have incorporated the feedback, which has been instrumental in making our manuscript clearer and more beneficial for the readership.

Thank you for guiding us in improving the quality of our work.

Please see below our responses to the comments:

More than 50 years have passed since the discovery of anthracyclines. Despite significant progress in the discovery of new anticancer drugs, anthracyclines are still highly effective drugs and are included in many cancer treatment regimens. It has long been known that anthracyclines exhibit precipitated cardiotoxicity, in some cases up to 40%.

Research is still ongoing to find molecules that would reduce the damaging effect of anthracyclines on the myocardium or prevent the initiation of programmed death of cardiomyocytes. A lot of drugs have been tested in experimental studies, but none of them has shown efficacy in clinical practice, and none of them is approved from the point of view of evidence-based medicine.

The article is relevant and opens new possibilities. At the same time, further research is needed.

There are no critical comments to the article.

It is advisable to include biomarkers (troponins, BNP) that are validated and are used as criteria for evaluating effectiveness.

Response: Thank you for your comment. You are correct that the inclusion of validated biomarkers such as troponins and B-type Natriuretic Peptide (BNP) is important in evaluating the effectiveness of treatments and interventions, particularly in a clinical context. This is indeed a histopathological study, and as such, the primary focus is on the microscopic examination of tissue samples to understand the manifestations of the disease at the cellular level. However, the integration of histopathological findings with clinical biomarkers can provide a more comprehensive understanding of the condition being studied. We will certainly consider investigating this further to enhance the scope and applicability of our research. Your input is very much appreciated.

Reviewer 2 Report

Comments and Suggestions for Authors

The study of Pop Moldovan et. Al. is a histopatological study designed to investigate the protective role of Hesperetin (HSP) in the context of epirubicin-triggered cardiomyopathy in mice. The study is significant for medicine as it examines the cardioprotective effects of HSP in response to a cumulative intraperitoneal dose of 12 mg/kg, which is considered a dose relevant to clinical settings.

Introduction:

"Their anticancer prowess stems from their ability to intercalate into nuclear DNA..." could be rephrased for flow. Consider: "Their anticancer efficacy is due in part to their ability to intercalate into nuclear DNA..."

Materials and Methods:
Ensure that the source for each reagent and piece of equipment is clearly identified.

Results:

1.Subheading "2.1. Hesperetin protect..." should be revised for grammatical correctness: "2.1. Hesperetin protects and maintains..."

2."Slightly increase in interstitial collagen fibers" should be "slight increase in interstitial collagen fibers."

3. "The group that received the combined treatment showed alleviate ultrastructural architecture" should be rephrased for clarity, perhaps to "The group that received the combined treatment showed alleviated ultrastructural damage."

Discussion:

1."Our findings reveal significant alterations in desmin distribution in the Epi-treated group compared to controls" - the phrase "compared to controls" could be placed at the end of the sentence for better readability.

2.The sentence structure throughout the discussion should be reviewed for clarity and conciseness.

Conclusions:

1."In summary, targeting the prevention of cardiomyocyte apoptosis presents a po-tential therapeutic strategy..." should be "In summary, targeting the prevention of cardiomyocyte apoptosis presents a potential therapeutic strategy..."

2."We propose that Hesperetin holds potential as a complementary treatment agent capable of averting the cardiomyopathy initiated by Epirubicin (Epi), whilst maintaining its anti-cancer efficacy." This is a strong statement that could benefit from more cautious language, such as "Our findings suggest that Hesperetin may serve as a complementary treatment..."

General Comments:

1.Ensure that all hyphenated words are correctly joined or separated as appropriate.

2.Check the consistency of terminology and formatting throughout the document (e.g., Epirubicin vs. EPI, hesperetin vs. HSP).

3.The authors should ensure that all figures and tables are properly referenced in the text.

4.The use of abbreviations should be consistent, and all should be defined at first use. For instance, if "HSP" is used as an abbreviation for hesperetin, it should be defined the first time it appears in the text.

Comments on the Quality of English Language

Minor editing of English language is required

Author Response

Dear Reviewer 2

We would like to express our sincere gratitude for the opportunity to submit our revised manuscript for your consideration. We deeply appreciate the insightful critiques and valuable suggestions provided. We have incorporated the feedback, which has been instrumental in making our manuscript clearer and more beneficial for the readership.

Thank you for guiding us in improving the quality of our work.

Please see below our responses to the comments:

The study of Pop Moldovan et. Al. is a histopatological study designed to investigate the protective role of Hesperetin (HSP) in the context of epirubicin-triggered cardiomyopathy in mice. The study is significant for medicine as it examines the cardioprotective effects of HSP in response to a cumulative intraperitoneal dose of 12 mg/kg, which is considered a dose relevant to clinical settings.

Introduction:

"Their anticancer prowess stems from their ability to intercalate into nuclear DNA..." could be rephrased for flow. Consider: "Their anticancer efficacy is due in part to their ability to intercalate into nuclear DNA..."

Response: Thank you for your sugestions. We change the phrase.

Materials and Methods:
Ensure that the source for each reagent and piece of equipment is clearly identified.

Response: We checked the entire section.

Results:

1.Subheading "2.1. Hesperetin protect..." should be revised for grammatical correctness: "2.1. Hesperetin protects and maintains..."

Response: we corrected

2."Slightly increase in interstitial collagen fibers" should be "slight increase in interstitial collagen fibers."

Response: we corrected

  1. "The group that received the combined treatment showed alleviate ultrastructural architecture" should be rephrased for clarity, perhaps to "The group that received the combined treatment showed alleviated ultrastructural damage."

 Response: we corrected

Discussion:

1."Our findings reveal significant alterations in desmin distribution in the Epi-treated group compared to controls" - the phrase "compared to controls" could be placed at the end of the sentence for better readability.

Response: we changed

2.The sentence structure throughout the discussion should be reviewed for clarity and conciseness.

 Response: we revised the entire text

Conclusions:

1."In summary, targeting the prevention of cardiomyocyte apoptosis presents a po-tential therapeutic strategy..." should be "In summary, targeting the prevention of cardiomyocyte apoptosis presents a potential therapeutic strategy..."

 Response: we changed

2."We propose that Hesperetin holds potential as a complementary treatment agent capable of averting the cardiomyopathy initiated by Epirubicin (Epi), whilst maintaining its anti-cancer efficacy." This is a strong statement that could benefit from more cautious language, such as "Our findings suggest that Hesperetin may serve as a complementary treatment..."

General Comments:

1.Ensure that all hyphenated words are correctly joined or separated as appropriate.

2.Check the consistency of terminology and formatting throughout the document (e.g., Epirubicin vs. EPI, hesperetin vs. HSP).

3.The authors should ensure that all figures and tables are properly referenced in the text.

4.The use of abbreviations should be consistent, and all should be defined at first use. For instance, if "HSP" is used as an abbreviation for hesperetin, it should be defined the first time it appears in the text.

Response: we checked all.

Reviewer 3 Report

Comments and Suggestions for Authors

In this study, authors described the cardioprotective effects of Hesperetin in a mouse model of Epirubicin-induced cardiomyopathy. Specifically, daily treatment (up to 13 days) with Hesperetin rescued cardiomyocytes from Epirubicin-induced myofibril loss, mitochondria damage, and apoptosis. The study examines an important clinical issue afflicting many cancer patients and survivors. Whilst it did demonstrate the therapeutic potential of Hesperetin, the study did not comprehensively explore and investigate the underlying cardioprotective mechanisms. Furthermore, this study also lacks cardiac function assessments. The following are comments and suggestions for the authors:

General Comments:

The introduction lacked a clear hypothesis and aims of the study. The rationale of the study was also indistinct. Can the authors clarify this?

Can the authors provide additional information on the treatment regime of the model? What is the rationale for the dosage of the Hesperetin (100 mg/kg)? Was this based on a dose response study or previous studies with murine models of cardiotoxicity? What is the rationale of using male CD1 mice?

Given that oxidative stress plays a critical role in anthracycline-induced cardiotoxicity, have the authors determined whether oxidative stress occurred in this model and Hesperetin was able to ameliorate Epirubicin-induced oxidative damage?

If Hesperetin is to be used in conjunction with Epirubicin to treat cancer, does Hesperetin has any effects on Epirubicin’s anticancer properties? Authors could examine this in relevant cancer cell lines.

Cardiac function assessments are the gold standard in detecting and diagnosing cardiotoxicity. Did the authors perform any cardiac function assessments? Did the authors examine the level of known cardiac biomarkers in plasma?

The discussion did elaborate on the limitation of this study such as the lack cardiac function assessments and molecular mechanisms. Was there any limitation in using Hesperetin (e.g., off-target effects)? What are the advantages of Hesperetin when compared to other antioxidants?  

Can the author speculate on how Hesperetin exert its cardioprotective effects on cardiomyocytes? What are the possible receptors and cellular pathways involved? Is this a direct or indirect signalling?

Specific Comments:

Page 5; Figure 1: The representative images of the FG and Desmin staining are not very clear in demonstrating the effects of Hesperetin. Can the authors provide better images at higher magnification? The image insert used to demonstrate TUNEL-positive cells in Figure 6 would be ideal.

Page 5; Figure 1: The Desmin staining images for the Control and HSP groups appeared to be from the same section. An artifact in the tissue appeared to be similar on both images. Can the authors clarify this?

Comments on the Quality of English Language

English language used in the manuscript was appropriate and understandable. Some figures and images required editing for better clarity.

Author Response

Dear Reviewer 3

We would like to express our sincere gratitude for the opportunity to submit our revised manuscript for your consideration. We deeply appreciate the insightful critiques and valuable suggestions provided. Each comment has been carefully considered, and we acknowledge the constructive role their thoughtful and critical assessments have played in enhancing our manuscript. We have incorporated the feedback, which has been instrumental in making our manuscript clearer and more beneficial for the readership.

Thank you for guiding us in improving the quality of our work.

Please see below our responses to the comments:

In this study, authors described the cardioprotective effects of Hesperetin in a mouse model of Epirubicin-induced cardiomyopathy. Specifically, daily treatment (up to 13 days) with Hesperetin rescued cardiomyocytes from Epirubicin-induced myofibril loss, mitochondria damage, and apoptosis. The study examines an important clinical issue afflicting many cancer patients and survivors. Whilst it did demonstrate the therapeutic potential of Hesperetin, the study did not comprehensively explore and investigate the underlying cardioprotective mechanisms. Furthermore, this study also lacks cardiac function assessments. The following are comments and suggestions for the authors:

General Comments:

The introduction lacked a clear hypothesis and aims of the study. The rationale of the study was also indistinct. Can the authors clarify this?

Response: Thank you for your insightful feedback regarding the clarity of our introduction. We added the hypothesis of this study

Can the authors provide additional information on the treatment regime of the model? What is the rationale for the dosage of the Hesperetin (100 mg/kg)? Was this based on a dose response study or previous studies with murine models of cardiotoxicity? What is the rationale of using male CD1 mice?

Response: The hesperetin (HSP) dosage of 100 mg/kg was chosen in light of dose-response studies and literature precedent demonstrating its efficacy in cardioprotective activity  (Agrawal et al., 2014).

We introduce this to the Material and Methods section.

The CD1 mice offer a robust model with less genetic variability compared to inbred strains, which allows for more generalizable findings. We use of a single sex—male in this case—was to prevent sex-based hormonal variations from affecting the study's results. The choice of male mice is consistent with the majority of preclinical studies in this field, facilitating comparisons with existing literature.

Given that oxidative stress plays a critical role in anthracycline-induced cardiotoxicity, have the authors determined whether oxidative stress occurred in this model and Hesperetin was able to ameliorate Epirubicin-induced oxidative damage?

Response: In our histopathological study, while we did not directly measure markers of oxidative stress, the protective effects of HSP observed at the administered dose are consistent with its well-documented antioxidant properties. The dosage chosen is based on previously established evidence where similar concentrations have demonstrated the ability to reduce oxidative stress in murine models (Kumar et al, 2013). Although the primary focus of our analyses was on histopathological changes and not on biochemical markers of oxidative stress, the reversal of Epi-induced myocardial damage and the preservation of cardiac structure following HSP treatment indirectly support the role of HSP in mitigating oxidative stress.The reduction in apoptosis and the preservation of mitochondrial integrity observed in our histopathological study upon HSP administration could potentially be attributed to its antioxidant capabilities, suggesting that these effects might underlie the cardioprotective benefits of HSP. This is underscored by the well-known link between mitochondrial dysfunction and oxidative stress.

If Hesperetin is to be used in conjunction with Epirubicin to treat cancer, does Hesperetin has any effects on Epirubicin’s anticancer properties? Authors could examine this in relevant cancer cell lines.

Response:  Thank you for your constructive query regarding the potential impact of hesperetin (HSP) on the anticancer properties of epirubicin (Epi). We acknowledge the importance of understanding whether HSP could influence Epi's effectiveness against cancer cells. The focus of our current study, however, was specifically to evaluate the cardioprotective effects of HSP against Epi-induced toxicity and not its impact on the chemotherapeutic efficacy of Epi. Hence, the interaction between HSP and Epi's anticancer actions was not within the scope of this investigation.

Nonetheless, we agree that your suggestion holds significant merit for a comprehensive evaluation of the combination therapy. We believe that conducting additional in vitro and in vivo studies to examine the effects of HSP on Epi's anticancer activity would be an intriguing and valuable extension of our research. Such studies could provide crucial insights into whether HSP preserves the therapeutic benefits of Epi while simultaneously protecting cardiac tissues.

Cardiac function assessments are the gold standard in detecting and diagnosing cardiotoxicity. Did the authors perform any cardiac function assessments? Did the authors examine the level of known cardiac biomarkers in plasma?

Response:Thank you for your comment emphasizing the importance of cardiac function assessments and biomarker analysis in the context of cardiotoxicity research.In the current study, our primary focus was on histopathological examination of cardiac tissue to assess the impact of hesperetin (HSP) on epirubicin (Epi)-induced structural damage in the heart. As such, we did not perform in vivo cardiac functional assessments or measure cardiac biomarkers in plasma, which are indeed the gold standards for diagnosing cardiotoxicity. However, we recognize the critical nature of these assessments in providing a comprehensive understanding of cardiac health and function. We fully intend to include such evaluations in a future, more extensive study that would allow for a more robust and thorough investigation.

The discussion did elaborate on the limitation of this study such as the lack cardiac function assessments and molecular mechanisms. Was there any limitation in using Hesperetin (e.g., off-target effects)? What are the advantages of Hesperetin when compared to other antioxidants?  

Can the author speculate on how Hesperetin exert its cardioprotective effects on cardiomyocytes? What are the possible receptors and cellular pathways involved? Is this a direct or indirect signalling?

 Response: Thank you for your insightful comments. The discussion in the manuscript was indeed designed to address the study's limitations, including the lack of direct cardiac function measurements and a detailed exploration of the molecular mechanisms involved.

Regarding the use of Hesperetin, while the study did not specifically focus on off-target effects. Future studies are planned to more thoroughly investigate the specificity of Hesperetin's action.

Specific Comments:

Page 5; Figure 1: The representative images of the FG and Desmin staining are not very clear in demonstrating the effects of Hesperetin. Can the authors provide better images at higher magnification? The image insert used to demonstrate TUNEL-positive cells in Figure 6 would be ideal.

Page 5; Figure 1: The Desmin staining images for the Control and HSP groups appeared to be from the same section. An artifact in the tissue appeared to be similar on both images. Can the authors clarify this?

Response: we changed the pannel for Desmin and FG with a higher magnification pictures.

Round 2

Reviewer 3 Report

Comments and Suggestions for Authors

The authors have addressed my feedback and suggestions in a satisfactory manner. They have improved the quality of the manuscript and enhanced its readership.

May I suggest to the authors to include a short discussion on the limitation/future direction of the study and incorporate their Response 4 and 5 into the discussion? These will highlight the significance for a comprehensive evaluating HSP as combination therapy in cancer and the importance of cardiac function assessments and biomarker analysis in the emerging field of Cardio-Oncology. 

Once this has been addressed, I recommend the manuscript to be accepted for publication.

Author Response

Dear Reviewer 3

We would like to express our sincere gratitude for the opportunity to submit our revised manuscript for your consideration. We deeply appreciate the insightful critiques and valuable suggestions provided. 

Thank you for guiding us in improving the quality of our work.

Please see below our responses to the comments:

May I suggest to the authors to include a short discussion on the limitation/future direction of the study and incorporate their Response 4 and 5 into the discussion? These will highlight the significance for a comprehensive evaluating HSP as combination therapy in cancer and the importance of cardiac function assessments and biomarker analysis in the emerging field of Cardio-Oncology. 

Response: Thank you for your valuable sugestions. We introduced at the conclusions as a limitation/future direction of the study.
